# A Small Molecule, 4-Phenylbutyric Acid, Suppresses HCV Replication via Epigenetically Induced Hepatic Hepcidin

**DOI:** 10.3390/ijms21155516

**Published:** 2020-08-01

**Authors:** Kiyoon Kim, Young-seok Lee, Suyun Jeong, Daehong Kim, Suk Chon, Youngmi Kim Pak, Sungsoo Kim, Joohun Ha, Insug Kang, Wonchae Choe

**Affiliations:** 1Department of Biochemistry and Molecular Biology, School of Medicine, Kyung Hee University, Seoul 02447, Korea; yoonkky@khu.ac.kr (K.K.); ykpak@khu.ac.kr (Y.K.P.); sgkim@khu.ac.kr (S.K.); hajh@khu.ac.kr (J.H.); iskang@khu.ac.kr (I.K.); 2Department of Biomedical Science, Graduate School, Kyung Hee University, Seoul 02447, Korea; wetoo123@gmail.com (Y.-s.L.); suyanjjang@naver.com (S.J.); 3Department of Radiology, National Cancer Center, Gyeonggi-do 10408, Korea; dkim@ncc.re.kr; 4Department of Endocrinology and Metabolism, School of Medicine, Kyung Hee University, Seoul 02447, Korea; imdrjs@naver.com; 5Department of Physiology, School of Medicine, Biomedical Science Institute, Kyung Hee University, Seoul 02447, Korea; 6Department of Biochemistry and Molecular Biology, Medical Research Center for Bioreaction to Reactive, Species and Biomedical Science Institute, School of Medicine, Kyung Hee University, Seoul 02447, Korea

**Keywords:** 4-phenylbutyric acid, hepatitis C virus, membranous web, histone deacetylase inhibitor, small molecule

## Abstract

Hepatic hepcidin is a well-known major iron regulator and has been reported to be closely related to hepatitis C virus (HCV) replication. However, pharmacological targeting of the hepcidin in HCV replication has not been reported. A short-chain fatty acid, 4-Phenyl butyrate (4-PBA), is an acid chemical chaperone that acts as a histone deacetylase inhibitor (HDACi) to promote chromosomal histone acetylation. Here, we investigated the therapeutic effect of 4-PBA on hepcidin expression and HCV replication. We used HCV genotype 1b Huh 7.5-Con1 replicon cells and engraftment of NOD/SCID mice as in vitro and in vivo models to test the effect of 4-PBA. It was found that 4-PBA inhibited HCV replication in Huh7.5-Con1 replicon cells in a concentration- and time-dependent manner through the induction of hepcidin expression by epigenetic modification and subsequent upregulation of interferon-α signaling. HCV formed a membranous web composed of double-membrane vesicles and was utilized for RNA replication. Moreover, 4-PBA also disrupted the integrity of the membranous web and interfered with the molecular interactions critical for the assembly of the HCV replication complex. These findings suggest that 4-PBA is a key epigenetic inducer of anti-HCV hepatic hepcidin and might at least in part play a role in targeting host factors related to HCV infection as an attractive complement to current HCV therapies.

## 1. Introduction

Approximately 3% of the global population is infected with Hepatitis C Virus (HCV) [1], which is a major cause of chronic liver diseases, such as steatosis, cirrhosis, and hepatocellular carcinoma (HCC). Genotype 1 is the most common genotype and is highly prevalent worldwide. Of the 53% of genotype 1 cases for which the subtype was specified, 99% were subtypes 1a or 1b (31% and 68%, respectively) [2]. Over the past few years, the number of patients successfully cured of hepatitis has increased drastically. In 2011, oral direct-acting antiviral reagents (DAAs), such as boceprevir and telaprevir was approval. At the end of 2013, a paradigm shift arose with the approval of simeprevir and sofosbuvir [3]. These drugs with or without PEG-IFN + RBV in selected genotypes were able to show an over 90% sustained virological response rate. However, there remained considerable limitations such as high DAA costs, DAA-resistant HCV variants and persistent HCC risks after cure [3]. Evidence from several cohorts suggests that HCC development and recurrence may be more frequent after DAA treatment compared with IFN-based therapy [4]. Therefore, targeting host factors required for virus infection might be an attractive complementary strategy to address these challenges.

4-PBA is a low-molecular-weight fatty acid and nontoxic pharmacological compound that has been approved by the United States Food and Drug Administration (FDA) for the treatment of urea cycle disorders [5]. This compound also acts as a HDACi in glioma and acute myeloid differentiation [6] and has been used to alleviate endoplasmic reticulum (ER) stress and related pathological conditions, including inflammation, hypertension, diabetes, cancers, and neurodegenerative disease. Further, 4-PBA reportedly induces cathelicidin antimicrobial peptides (AMPs), we hypothesized that 4-PBA might induce hepcidin expression [7]. However, the effects of 4-PBA on HCV RNA replication are unknown.

Butyrate is mainly present in the gastrointestinal tract and produced by gut bacterial fermentation. It acts as a messenger between the microbiota and the immune system [8,9]. As an odorless derivative of butyrate, 4-PBA has been reported to induce immune responses through cathelicidin AMP induction in several cell lines [7]. Hepcidin is a member of the AMP family, and its expression regulated by iron status, hypoxia, oxidative stress, inflammation, and infection. Hepcidin is a liver peptide hormone encoded by the *HAMP* gene and acts as a key regulator of iron entry into the circulation. When hepcidin levels are abnormally high, serum iron levels are reduced owing to iron trapping in macrophages and liver cells. When hepcidin levels are abnormally low, iron overload occurs [10,11]. Moreover, this highly disulfide-bonded peptide hormone has antimicrobial activity against bacteria and fungi. Recently, hepatic hepcidin was shown to be closely related to the regulation of HCV replication and chronic hepatitis C. HCV infection inhibits hepcidin expression, and treatment with synthetic hepcidin peptide has broad anti-HCV activity, at least for genotypes 1 and 2. These effects are related to the activation of IFN-inducible genes through increased hepcidin expression. However, the exact mechanisms are still unclear [12].

Accordingly, in this study, we evaluated the effects of 4-PBA on HCV replication. Our results provided insights into the mechanisms of HCV RNA replication through the induction of hepcidin expression.

## 2. Results

### 2.1. 4-PBA Inhibited HCV RNA Replication

To investigate the effects of 4-PBA on HCV RNA replication, we first quantified HCV RNA by qRT-PCR and luciferase assays in genotype 1b Huh 7.5-Con1 replicon cells. The structure of 4-PBA has showed in Figure 1a. It was found that 4-PBA decreased HCV RNA levels relative to that in untreated cells (Figure 1b,c). To confirm the inhibition of HCV RNA replication by 4-PBA, we examined the expression of NS5A and NS3, two viral proteins essential for HCV replication, by immunoblotting. NS3 protein levels were downregulated by treatment with 4-PBA (Figure 1d). NS5A expression was not significantly altered by treatment with 4-PBA for 24 h, although significant decreases were observed after 48 and 72 h. The downregulation of NS5A after 72 h of treatment with 4-PBA was confirmed by immunofluorescence analysis (Figure 1e).

Because 4-PBA is known to protect cells by alleviating ER stress, we examined whether 4-PBA inhibited HCV replication by reducing ER stress. We treated HCV replicon-containing Huh7.5-Con1 cells with 1 mM 4-PBA and performed immunoblotting to monitor the levels of the ER stress markers grp78, protein kinase R-like ER kinase, and inositol-requiring transmembrane kinase/endoribonuclease 1α. The 4-PBA concentration used in our study (1 mM) did not alleviate ER stress in Huh7.5-Con1 cells (Appendix A). In addition, 4-PBA had no effect on cell viability after 24 and 48 h of treatment at concentrations up to 2 mM, compared with that in untreated cells (Appendix A). These data suggested that 4-PBA inhibited HCV replication without its cytotoxicity, independent of HCV-induced ER stress.

### 2.2. 4-PBA Induced the Expression of the Antimicrobial Peptide (AMP), Hepcidin

Recently, several studies reported that hepcidin expression is reduced by HCV replication-induced oxidative stress and that hepcidin peptide treatment inhibits HCV replication [13]. Based on these previous studies, we speculated that 4-PBA may inhibit HCV replication by regulation of host AMP expression. To test this idea, we examined the expression levels of hepcidin after treatment with 4-PBA by immunofluorescence staining (Figure 2a) and qRT-PCR (Figure 2b). As expected, 4-PBA induced mRNA and peptide expression of hepcidin. We also performed immunohistochemistry using the livers of vehicle- and 4-PBA-treated mice. Hepatic hepcidin was highly expressed in 4-PBA-treated tissues compared with that in vehicle-treated tissues (vehicle: 0.07; 75 mg/kg: 0.21; 150 mg/kg: 0.32; Figure 2c,d).

Next, we investigated whether hepcidin induction was associated with epigenetic modification at the hepcidin locus. Chromatin immunoprecipitation (ChIP) qPCR assays were performed with genomic DNA from Huh7.5-Con1 cells treated with 4-PBA for 6, 12, or 24 h. 4-PBA-induced histone acetylation at H3K9 (Figure 3a) and H3K27 (Figure 3b) in a time-dependent manner. In particular, histone acetylation at H3K27 was increased by 2–3-fold after 4-PBA treatment. We also found that 4-PBA treatment increased RNA polymerase II binding at the hepcidin promoter, confirming that hepcidin induction was mediated by transcriptional initiation (Figure 3c). The treatment of Huh7.5-Con1 cells with 4-PBA also increased histone acetylation at H3K27 in whole-cell lysates (Figure 3d). These data indicated that 4-PBA treatment enhanced hepcidin expression through epigenetic modification at the hepcidin locus.

### 2.3. 4-PBA Enhanced the IFN-α Response

During HCV replication, hepcidin peptide treatment stimulates the expression of IFN-induced genes [12]. To monitor the effects of 4-PBA-induced hepcidin on host immune responses, we first determined the activity of the IFN-stimulated response element (ISRE) after treatment with 4-PBA using luciferase assay. Our results showed that 4-PBA treatment increased ISRE activity by about three-fold (Figure 4a). Next, we analyzed IFN-α mRNA and protein expression in Huh7.3-Con1 cells treated with 4-PBA. We observed that 4-PBA treatment induced IFN-α mRNA expression (Figure 4b, left panel) as well as IFN-α production and secretion, compared with that in untreated cells (Figure 4b, right panel). Moreover, 4-PBA treatment also increased mRNA expression of IFN regulatory factors (IRFs; Figure 4c) and of IFN-α-inducible innate immune response-related antiviral genes, such as protein kinase R (PKR), IFN-induced protein with tetratricopeptide repeats 1 (IFIT1), and 2′-5′-oligoadenylate synthetase 1 (OAS1; Figure 4d).

### 2.4. Hepcidin Mediated 4-PBA-Induced IFN-α Signaling

To further explore the role of hepcidin in 4-PBA-induced IFN-α signaling, we performed RNA interference experiments using small interfering RNAs (siRNAs) specific for hepcidin. Hepcidin siRNA (si*HAMP*) transfection successfully resulted in reduced hepcidin mRNA expression (Figure 5a). We then evaluated the expression, production, and regulation of IFN-α, as in Figure 4b,c, with hepcidin knockdown. As shown in Figure 5b, in hepcidin-knockdown cells, 4-PBA-induced IFN-α expression was blocked compared with that in control siRNA (siCON)-transfected cells (Figure 5b). Moreover, hepcidin knockdown also decreased the mRNA expression of IRF1, IRF3, and IRF7 after 4-PBA treatment (Figure 5c). As shown in Figure 5d, after treatment with 4-PBA, hepcidin knockdown significantly reduced the mRNA expression of IFN-α-inducible genes, such as PKR, IFIT1, and OAS1, compared with that in siCON-transfected cells. Finally, we analyzed HCV RNA replication in hepcidin-knockdown cells by qRT-PCR (Figure 5e) and luciferase assays (Figure 5f). Compared with siCON-transfected cells, hepcidin-knockdown cells showed suppression of 4-PBA-dependent reduction in HCV RNA replication. Interestingly, compared to siCON, HCV RNA replication was increased independently of 4-PBA by about three-fold in hepcidin-knockdown cells. Altogether, our results suggested that 4-PBA treatment inhibited HCV RNA replication by increasing hepcidin expression and IFN-α signaling.

### 2.5. 4-PBA Disrupted MW Formation

HCV replication and viral protein expression occur in specialized structures, called the membranous web (MW). The MW protects the viral genome from host immune responses by limiting its interactions with cytosolic pattern-recognition receptors [14,15]. Although we have successfully shown that 4-PBA inhibits HCV replication, we wanted to directly confirm MW disruption by 4-PBA treatment.

Upon loss of MW integrity, aberrantly recruited NS5A proteins form a cluster [16]. As shown in Figure 1d, treatment with 4-PBA for 24 h did not significantly reduce NS5A expression levels. However, further analysis revealed increasing abnormal NS5A clustering compared with that in untreated cells. Therefore, in order to compare the observed NS5A clustering with that of a known NS5A inhibitor, DAV, immunofluorescence analysis was performed again after treatment with 4-PBA for 24 h. Our results revealed that 4-PBA treatment augmented abnormal NS5A clustering by ~2.5 fold, similar to cells treated with the NS5A inhibitor DAV (Figure 6a,b). Because replication complex proteins located in the MW are covered with lipid droplets, a substantial proportion of NS5A protein is associated with detergent-resistant insoluble membrane fractions [17]. We therefore monitored the distribution of NS5A protein in lipid droplets with membrane flotation assays. Cell lysates incubated with or without mild detergent were separated on 65%, 55%, and 10% discontinuous sucrose gradients. Insoluble membrane proteins and lipid droplets were located between the 55% and 10% sucrose layers, whereas cytosol-solubilized membrane proteins remained in the 62% sucrose layer [17]. As expected, a large proportion of NS5A proteins were found to be associated with the detergent-resistant membrane fraction. After incubation with mild detergent, only a small proportion of the proteins remained in the detergent-resistant membrane fraction (Figure 6c, fractions 1–4). However, a significant amount of NS5A in 4-PBA-treated cell lysates was present in the solubilized membrane protein fraction, even without mild detergent treatment (Figure 6c, fractions 5–9). These data indicated that 4-PBA interfered with MW formation and translocation of NS5A protein to the solubilized membrane protein fraction.

### 2.6. 4-PBA Inhibited the Formation of Double Membranous Vesicles (DMVs)

We next studied the effects of 4-PBA on MW formation by electron microscopy. HCV-induced MWs serving as viral replication sites consist primarily of DMVs. In accordance with previous reports [16,18,19], HCV-induced DMVs (104.82 ± 29.43 nm) were observed in Huh7.5-Con1 cells but not in mock-infected cells (Figure 7a,b). Treatment with 4-PBA markedly reduced the number and size of DMVs (Figure 7b,c). After 24 h of treatment with 4-PBA, there were almost no DMVs (Figure 7a,c), and those that remained were smaller in size than untreated DMVs (35.87 ± 12.68 versus 104.82 ± 29.43 nm; Figure 7b). Similar results were obtained following treatment with DAV (Figure 7b,c). Interestingly, 4-PBA and DAV-treated cells had similar multimembranous vesicle (MMV) structures (Appendix A.). MMVs appeared at late stages of infection as a result of stress or autophagy [18]. The results of our electron microscopy analysis suggested that 4-PBA interfered with HCV replication by preventing DMV formation.

### 2.7. 4-PBA Interfered with Molecular Interactions in the HCV RNA Replication Complex

Previous studies have shown that the molecular interactions among VAMP-associated protein A (hVAP-33), NS5A, and NS5B are critical for the assembly of the HCV RNA replication complex [20]. Therefore, we examined interactions of oxysterol-binding protein 1 (OSBP), VAMP associated protein B (VAP-B), hVAP-33, and cyclophilin (Cyp) B with the HCV replication complex proteins NS5A and NS5B in lysates of Huh7.5-Con1 cells treated with 4-PBA for 24 h by immunoprecipitation. We found that 4-PBA treatment altered the interactions between NS5A and other host cellular components (Figure 8a–c); the interactions between NS5A and OSBP and between NS5A and VAP-B were reduced by almost 50%, whereas the interaction between NS5A and hVAP-33, which is critical for replication of HCV RNA, was decreased by almost 80% (Figure 8c). In contrast, 4-PBA treatment enhanced the interaction between NS5B and hVAP-33 (Figure 8d). Immunoprecipitation with anti-NS5A antibodies showed that 4-PBA suppressed the interaction between NS5A and NS5B by almost 50% (Figure 8e). We also performed immunoprecipitation with an antibody against CypB, a molecular chaperone that functions as a stimulatory cofactor for NS5B in the replication complex [21]. However, 4-PBA did not affect the interaction between NS5B and CypB (Figure 8f). To rule out the possibility that the decrease in the NS5A/hVAP-33 interaction was related to downregulation of hVAP-33, we monitored the expression levels of the proteins (Appendix A) and mRNAs (Appendix A) after 4-PBA treatment and found that hVAP-33 expression was unchanged. Pull-down assays using HEK293T cells confirmed that 4-PBA treatment inhibited the interaction between NS5A and hVAP-33 but increased that between NS5B and hVAP-33 (Appendix A). Thus, we concluded that 4-PBA may alter protein/protein interactions in the replication complex, particularly those among hVAP-33, NS5A, and NS5B.

To validate our findings, we performed immunofluorescence analysis of NS5A, NS5B, and hVAP-33 expression in cells treated with 4-PBA. Consistent with previous reports [17], viral NS5A and endogenous hVAP-33 colocalized in the absence of 4-PBA (Figure 8g, left panel). However, this colocalization was abolished by incubation with 4-PBA for 24 h (Figure 8g, right panel). Interestingly, treatment with 4-PBA increased NS5B and hVAP-33 colocalization relative to that in untreated cells (Figure 8h).

Cumulatively, these results suggested that 4-PBA treatment disrupted replication complex interactions.

### 2.8. The Anti-HCV Effects of 4-PBA in a Mouse Model

To evaluate the effects of 4-PBA on HCV inhibition in vivo, we generated an HCV replicon-based mouse model by engrafting Huh7.5-Con1 cells into immunodeficient NOD/SCID mice. Liver morphology was examined by magnetic resonance imaging (MRI) and anatomical observations at three and seven weeks after the surgery to confirm that the model was established (Figure 9a,b). Daily injection of 4-PBA (vehicle + 75 or 150 mg/kg) for seven days was initiated seven weeks after engraftment of HCV replicon cells. The antiviral effects of 4-PBA were evaluated by immunohistochemistry and immunofluorescence labeling of HCV NS5A, and engrafted Huh7.5-Con1 cells were identified based on the expression of the human hepatocyte-specific antigen OCH1E5. There was no significant loss of body weight related to the potential cytotoxicity of 4-PBA (Appendix A). However, 4-PBA-treated mice showed a significant reduction in NS5A protein levels in the OCH1E5-positive area as compared with that in vehicle-injected mice (Figure 9c,d). These findings provide strong evidence that 4-PBA inhibited HCV replication in vivo.

## 3. Discussion

The FDA-approved drug 4-PBA is currently used to treat several urea cycle disorders [5,22]. In this study, we found that 4-PBA, at clinically feasible concentrations [2,23], not only inhibited HCV replication through hepatic hepcidin-induced IFN-α signaling, but also disrupted the HCV replication complex and MW formation, providing strong evidence for the novel antiviral roles of 4-PBA in HCV RNA replication.

Our results showed that 4-PBA suppressed HCV replication in a concentration- and time- dependent manner. According to the FDA document of 4-PBA, average 11.45 g/m^2^/day (roughly equivalent to 5.1 mM for in vitro use) can be clinically used for urea cycle disorder treatment (based on patients over 20 kg). In this study, we used much lower concertation of 4-PBA (1 mM) to observe significant inhibitory effects on HCV RNA replication in vitro. Interestingly, 4-PBA treatment for 24 h did not alter NS5A protein levels, consistent with recent findings by other groups. For example, several NS5A inhibitors have been shown to block the assembly of new replication complexes, although they are ineffective against those that were already formed [24]. The DAV-like NS5A inhibitor BMS-553 was found to disrupt HCV replication complex formation, whereas NS5A abundance was not unaffected by 20-h treatment, consistent with our findings [25]. However, our immunohistochemical analysis confirmed that 4-PBA treatment for 72 h clearly reduced NS5A protein levels in Huh 7.5-Con1 cells.

A few studies reported that high dosage (4–10 mM) of 4-PBA was used to reduce ER stress in HCV study [26]. However, for the first time, we demonstrated that HCV inhibitory mechanism of 4-PBA treatment is based on epigenetic hepcidin regulation at much lower concentrations (1 mM), not related to ER stress reduction. Moreover, 4-PBA possesses HDACi activity, which contributes to alterations in gene expression patterns caused by epigenetic changes in chromatin structure. Although the mechanisms of action of HDACis are still unknown, these compounds are considered potential therapeutic agents in many diseases, including Hepatitis C. 4-PBA suppresses cell proliferation via p21 upregulation through its HDACi activity [27]. Moreover, 4-PBA induces *Abcd2* gene expression in glial cells in the same manner [28]. In this study, for the first time, we showed that 4-PBA induced hepcidin expression. Additionally, we found that 4-PBA induced histone acetylation in a time-dependent manner at hepcidin loci. Consistent with our data, HCV-induced oxidative stress has been reported to suppress hepcidin expression via HDAC activity, and trichostatin A, a known HDAC inhibitor, partially recovers hepcidin expression owing to its cytotoxicity at high concentrations [29]. Interestingly, in this study, histone acetylation at H3K27 was strongly induced after treatment with 4-PBA. Although we clearly showed that 4-PBA induced hepcidin expression by epigenetic modification at the hepcidin locus, more detailed mechanisms are needed. For example, it would be interesting to address which class of HDACs is affected in HCV replication following treatment with 4-PBA because 4-PBA is a class I and II HDACi.

Hepcidin plays important roles not only in iron homeostasis regulation but also as an antimicrobial and antiviral factor. Liu et al. reported that hepcidin inhibits HCV replication and increases signal transducer and activator of transcription 3 phosphorylation, resulting in increased expression of IFN-inducible genes [12]. In this study, we showed that 4-PBA activated the ISRE and increased IFN-α expression through hepcidin induction. Moreover, 4-PBA increased the expression of *IRF1, IRF3*, and *IRF7* and upregulated IFN-α-inducible innate immune response-related antiviral genes, such as *PKR*, *IFIT1*, and *OAS1*. Knockdown of hepcidin blocked the effects of 4-PBA on IFN-α signaling. In particular, *OAS1* mRNA expression induced by 4-PBA was completely dependent on hepcidin expression, suggesting that hepcidin was essential for the regulation of anti-HCV activity. Similarly, 4-PBA was unable to inhibit HCV replication in hepcidin-knockdown cells. Taken together, our findings suggested that 4-PBA inhibited HCV replication via the hepcidin-induced IFN-α response.

Cholesterol-rich DMWs composed of a double membrane are formed for HCV RNA replication [22,30]. This structure not only provides a basis for the interaction of replication complex components but also protects viral RNA from innate immune responses of the host [18,19]. The replication factory is disrupted by the inhibition of phosphatidylinositol 4-kinase and OSBP, resulting in increased formation of abnormal NS5A clusters, a phenomenon that was observed upon treatment with an NS5A inhibitor [31]. Therefore, we wanted to confirm that 4-PBA affects the stability of DMWs. When HCV replication was inhibited, DMVs shrank and eventually disappeared [25]. Consistent with previous reports, we found that 4-PBA treatment abrogated the DMWs of HCV. Moreover, we confirmed that the sizes and ratios of DMVs after 4-PBA treatment were reduced. We observed a few MMVs in cells treated with 4-PBA or DAV that were much larger than DMVs. Some MMVs have an electron-dense lumen that likely corresponds to engulfed cytosol, whereas others appeared empty; these may be double-membrane autophagosomes that had engulfed DMVs and the cytosol. In HCV replication kinetics, MMVs appear in late phages after exponential RNA synthesis and are therefore only of minor importance to viral replication [18]. The MW is located on lipid rafts, which are detergent-resistant membranes; thus, most replication complex-associated proteins are present in the insoluble fraction. Detergent-resistant membranes are most likely intracellular lipid rafts derived from the ER, Golgi, or trans-Golgi network [17]. Viral NS proteins are thought to be associated with ER membranes and recruited to lipid rafts only when participating in RNA replication [30,32]. In this study, we demonstrated that NS5A was no longer present in the insoluble fraction following treatment with 4-PBA. The mechanism through which 4-PBA disrupted DMWs has not been clearly demonstrated. It is plausible that 4-PBA treatment inhibits HCV RNA replication, eventually resulting in DMW disruption, and vice versa. No reports have described the role of hepcidin in DMW disruption dependently or independent with HCV RNA replication. Moreover, if 4-PBA has a role in DMW disruption independently with HCV RNA replication, it might hinder us from analyzing 4-PBA-dependent DMW disruption in a hepcidin-knockdown cell. Therefore, further studies using HCV polyprotein expression system are required to clarify the roles of 4-PBA in DMW disruption.

We showed that 4-PBA inhibited molecular interactions in the HCV RNA replication complex. Notably, protein/protein interactions between NS5A and hVAP-33 were inhibited by treatment with 4-PBA, whereas the interaction between NS5B and hVAP-33 was enhanced. NS5A and NS5B bind to different domains of hVAP-33, and NS5A can associate with hVAP-33 independent of NS5B [20]. In addition, suppression of HCV replication by OSBP-related protein 4 is linked to increased interactions between NS5B and hVAP-33, consistent with our findings. Thus, enhanced hVAP-33/NS5B binding in the absence of NS5A may have inhibitory effects on HCV replication. Finally, we confirmed the antiviral effects of 4-PBA in an HCV mouse model that persisted for one week after treatment.

Many patients affected by HCC have an impaired iron metabolism, showing low serum hepcidin levels in HCC tissue as compared to the non-neoplastic one [33]. Considering hypermethylation at *HAMP* promoter in HCC, co-treatment with 4-PBA and DAAs also might be beneficial to possibly reduce the HCC risk while treating chronic hepatitis.

## 4. Materials and Methods

### 4.1. Cell Culture

HCV Genotype 1b Huh7.5-Con1 replicon cells were provided by Apath (Apath, New York, NY, USA). Cells were grown in Dulbecco’s modified Eagle’s medium (DMEM) supplemented with 2 mM nonessential amino acids, 100 U/mL penicillin, 10% fetal bovine serum, and 300 μg/mL geneticin (Thermo Fisher Scientific, Waltham, MA, USA), in a humidified atmosphere (5% CO_2_, 37 °C).

### 4.2. Reagents

Breifly, 4-PBA and 4′,6-diamidino-2-phenylindole (DAPI) were purchased from Sigma-Aldrich (Sigma-Aldrich, St. Louis, MO, USA). Daclatasvir (DAV) was purchased from MedChem Express (MedChem Express, Monmouth Junction, NJ, USA). Antibodies used in this study include mouse anti-NS5A; sc-65458, mouse anti-NS3; sc-69938, mouse anti-a-actinin; sc-17829 rabbit anti-hVAP-33; sc-98890, anti-grp78;sc-13968, rabbit anti-PERK; sc-13073, rabbit anti-IRE1a; sc-20790, mouse anti-actin; sc-8432 (Santa cruz, Dallas, TX, USA), rabbit anti-hepcidin; ab75883, rabbit anti-H3K9ac; ab10812, rabbit anti-H3K27ac; ab4729, mouse anti-RNA pol III; ab817, rabbit anti-CypB; ab16045, (abCAM, Canbridge, UK), rabbit anti-OSBP; #11096-1-AP (proteintech, Rosemont, IL, USA), rabbit anti-VAP-B; NBP1-89112 (NOVUS, Centennial, CO, USA), mouse anti-OCH1E5; MBS370077, rabbit anti-NS5A (For Immunohistochemistry); MBS485095 (mybiosource, San Diego, CA, USA). Secondary antibodies used are goat anti-mouse IgG; M32607, goat anti-rabbit IgG; 31460, Alexa Fluor 488 goat anti-rabbit IgG; A-11034, Alexa Fluor 488 goat anti-mouse IgG; A-11001, Alexa Fluor 594 goat anti-mouse IgG; A-11005, Alexa Flour 594 goat anti-rabbit IgG; A-11012 (Invitrogen, Waltham, MA, USA).

### 4.3. Quantitative Real-Time Reverse Transcription Polymerase Chain Reaction (qRT-PCR)

qRT-PCR was performed using Power SYBR Green PCR Master Mix on an ABI Prism 7300 real-time PCR system (Applied Biosystems, Foster City, CA, USA). The integrity of the amplified DNA was confirmed by determining the melting temperature. Expression levels were normalized to that of β-actin. The forward and reverse primer sequences are listed in Table 1.

### 4.4. Determination of Cell Viability 

Cell viability was assessed by lactate dehydrogenase (LDH)-release assays. LDH-release assays (Promega, Medison, WI, USA) were performed under the same conditions according to the manufacturers’ instructions. Each value represents the mean of a minimum of nine wells.

### 4.5. Chromatin Immunoprecipitation Assay 

The chromatin immunoprecipitation (ChIP) assay was performed according to simpleChIP^R^Enzymatic Chromatin IP kit protocol (cell signaling, Danvers, MA, USA).

### 4.6. Membrane Flotation Assay

The membrane floatation assay was performed as previously described [17], with some modifications. Cells were lysed in 1 mL of hypotonic buffer composed of 10 mM Tris-HCl (pH 7.5), 10 mM KCl, and 5 mM MgCl_2_ and passed 20 times through a 25-G needle. The nuclei and unbroken cells were removed by centrifugation at 3000 rpm for 5 min at 4 °C. The cell lysates were mixed with 7 mL of 72% sucrose in low-salt buffer (LSB) composed of 50 mM Tris-HCl, 25 mM KCl, and 5 mM MgCl_2_, and overlaid with 7 mL of 55% and 3 mL of 10% sucrose-LSB solution. Each 2-mL fraction was collected from the top of the gradient after gradient centrifugation at 36,000 rpm in a SW32Ti rotor for 24 h (Beckman Coulter, Indianapolis, IN, USA) at 4 °C. Each fraction was concentrated by trichloroacetic acid protein precipitation, and protein pellets were resuspended in SDS sample buffer and analyzed by SDS-PAGE.

### 4.7. Luciferase Assay

The luciferase assay was performed using the Renilla Luciferase Assay System kit (Promega, Madison, WI, USA) according to the manufacturer’s instructions. 

### 4.8. Immunoblotting

Cells were lysed with lysis buffer (50 mM Tris-Cl, 150 mM NaCl, 5 mM EDTA, 0.1% NP-40) containing protease inhibitor cocktail (Calbiochem, San Diego, CA, USA) and sodium vanadate (NaVO_4_). Whole lysates were loaded into 10–14% SDS-PAGE gels for electrophoresis and transferred to nitrocellulose membrane. Membranes were blocked with 5% BSA for 1 h, washed, and incubated with primary antibody overnight at 4 °C. Following 45 min incubation with secondary antibody, proteins were visualized by enhanced Western Blotting Luminol Reagent (Santa Cruz, Dallas, TX, USA).

### 4.9. Immunofluorescence Analysis

Cells were seeded on coverslips in 24-well plates and were treated as indicated 1 day later. After culturing, the cells were fixed with 4% paraformaldehyde for 1 h at room temperature and permeabilized with 0.1%(*v*/*v*) NP-40 for 30 min. Samples were washed three times with phosphate-buffered saline (PBS) with 0.1%(*v*/*v*) Triton X-100 and incubated with 1%(*w*/*v*) BSA in PBS with 0.1%(*v*/*v*) Triton X-100. After blocking, the samples were incubated overnight at 4 °C with primary antibodies. After three washes, the cells were incubated with Alexa Fluor-488 or -546-conjugated secondary antibodies for 1 h at room temperature while protected from light.

### 4.10. Immunoprecipitation

Treated or co-transfected cells were lysed with lysis buffer, and cell lysates were cleared by centrifugation at 14,000 rpm for 15 min. The lysates (2 mg) were incubated with anti-hVAP-33 or -NS5A antibody (10 μg) for 12 h at 4 °C. Protein-A/G-conjugated agarose beads were added, followed by incubation for 5 h at 4 °C. Beads were washed three times with 1× TBST. Immunopellets were boiled with SDS-PAGE sample buffer and resolved by electrophoresis.

### 4.11. Immunohistochemistry

Immunolabeling was performed using a Dako REAL EnVision Detection System (Agilent Technologies, San Diego, CA, USA). For immunohistochemical detection of NS5A and OCH1E5, samples were incubated overnight at 4 °C with monoclonal antibodies against NS5A (1:500). For fluorescence immunohistochemistry, deparaffinized and rehydrated liver tissue sections were incubated for 16 h at 4 °C with anti-NS5A and -OCH1E5 antibodies. Specimens were mounted in Fluoromount Aqueous Mounting Medium (Sigma-Aldrich, St. Louis, MO, USA) and visualized by confocal microscopy.

### 4.12. DNAs and siRNAs Transfection

Plasmids and siRNAs were transfected using Xtreamgene HP according to manufacturers’ instructions (Roche, Swiss). Hepcidin-specific siRNA and scrambled siRNA were purchased from Dharmacon (Dharmacon, Denver, CO, USA) and transfected using Viromer BLUE according to the manufacturers’ instructions (lipocalyx, Germany)

### 4.13. Electron Microscopy

Huh7.5-Con1 replicon cells were treated with 1 mM 4-PBA or 2 μM DAV for 24 h, then fixed in 2.5% glutaraldehyde in phosphate buffer (pH 7.3). After one time rinses in buffer, the cells were post-fixed with 1% osmium tetroxide in the same buffer and stained en bloc with an aqueous, saturated solution of uranyl acetate. After rapid dehydration in a graded ethanol series, the samples were infiltrated with and embedded in Epon. Ultra-thin sections were cut and collected onto copper grids and stained with uranyl acetate and lead citrate, and viewed on a Tecnai G^2^ Sprit Twin electron microscope (FEI, Hillsboro, OR, USA) at 120 kV. Images were digitally captured with a US4000 charge-coupled device camera (Gatan, Pleasanton, CA, USA) using digital micrograph software (Gatan).

### 4.14. Animal Procedures

This study was reviewed and approved by the Institutional Animal Care and Use Committee (IACUC) of National Cancer Center Research Institute. NCCRI is an Association for Assessment and Accreditation of Laboratory Animal Care International (AAALAC International) accredited facility and abide by the Institute of Laboratory Animal Resources (ILAR) guide. (National Cancer Center, Goyang, South Korea; approval ID No. NCC-15-249, approval date: 23 April 2015). An HCV mouse model was established as previously described [34]. Briefly, male immunodeficient NOD/SCID mice (6 weeks old; Charles River Laboratories, Wilmington, MA, USA) were engrafted with HCV replicon-containing Huh7.5-Con1 cells by intrasplenic injection. Cell transplantation and surgical procedures were performed under anesthesia. At 3 weeks after cell transplantation, engraftment of the cells was monitored with MRI scanning. At 7 weeks after cell transplantation, dimethyl sulfoxide or 4-PBA was dissolved in polyethylene glycol and administered daily to the mice by intraperitoneal injection. During the treatment, body weights were measured every 2 days. Each group consisted of five animals. At 7 days after the last injection, the mice were sacrificed, and their liver tissues were collected for analysis. Surgery and injection were carried out, and animals were allowed to recover in a specific pathogen-free room. Mice were fed a normal diet and housed with corncob bedding in specific pathogen-free conditions.

### 4.15. Data and Statistical Analysis

Results are expressed as means ± SEM of five independent experiments. Depending on the design of the experiment, data were analyzed with Student’s *t*-tests, one-way or two-way ANOVA. When ANOVA achieved the necessary level of statistical significance (*P* < 0.05) and when no significant variance inhomogeneity was detected, post hoc multi-comparison analysis (Tukey’s test) was applied. Statistical analysis was performed using GraphPad Prism v.8.2.1. Differences were considered statistically significant when *P* < 0.05. Group sizes were designed to be equal and always represent the number of experimental independent repeats, each carried out with distinct biological preparations.

## 5. Conclusions

In conclusion, we demonstrated that the short-chain fatty acid 4-PBA inhibited HCV replication in vitro and in vivo. Moreover, 4-PBA increased hepcidin expression via H3K27 acetylation, thereby inducing IFN-α signaling. Meanwhile, 4-PBA also abrogated MW formation and disrupted the molecular interactions of replication complex components. Further studies on the antiviral activity of 4-PBA could establish this compound for targeting host factors related to HCV infection as an attractive complement to current HCV therapies.

## Figures and Tables

**Figure 1 ijms-21-05516-f001:**
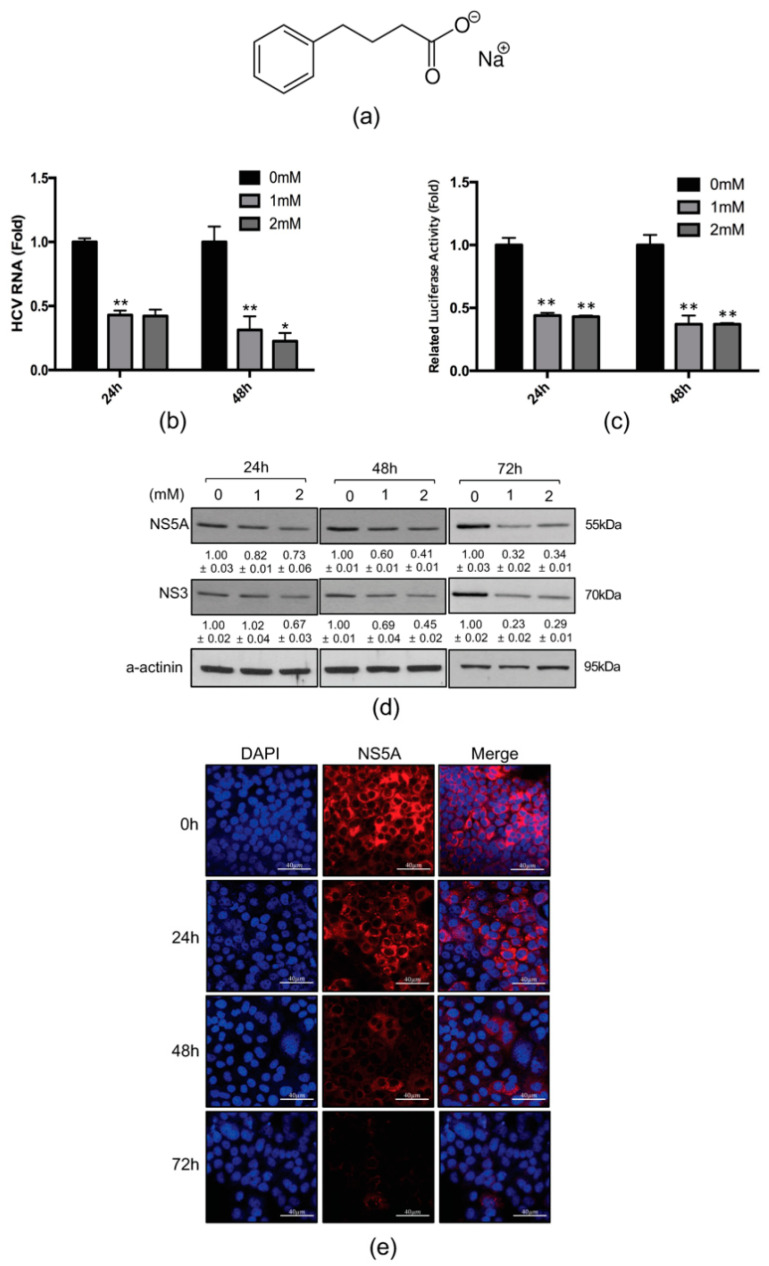
4-PBA inhibited HCV 1b RNA replication. (**a**) A chemical structure of 4-PBA. Huh7.5-Con1 cells were treated with 4-PBA at the indicated doses for 24 or 48 h. HCV replication was measured by (**b**) qRT-PCR and (**c**) luciferase assays. (**d**) Expression of HCV viral proteins was analyzed by immunoblotting. Whole lysates from cells with or without 4-PBA treatment were subjected to western blotting using anti-NS3 and anti-NS5A antibodies; α-actinin was used as a loading control. (**e**) Huh7.5-Con1 cells treated with or without 1 mM 4-PBA were stained for NS5A (**red**) expression, and DAPI was used for nuclear counterstaining (**blue**). Scale bar = 40 μm Data are expressed as means ± SDs of five independent experiments. * *P* < 0.05, ** *P* < 0.01 versus untreated control.

**Figure 2 ijms-21-05516-f002:**
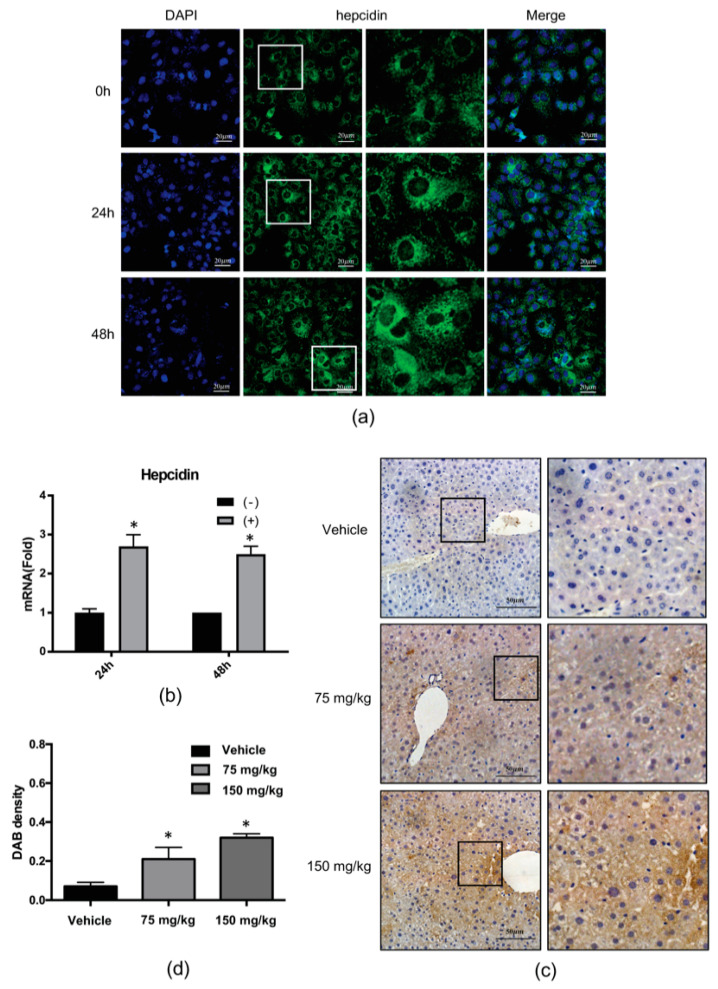
4-PBA induced hepatic hepcidin expression. (**a**) Indirect immunofluorescence analysis of Huh7.5-Con1 cells with or without 1 mM 4-PBA treatment for 24 or 48 h using antibodies against hepcidin (**green**). DAPI (blue) was used to visualize nuclei. Scale bar = 20 μm. (**b**) After treatment with 1 mM 4-PBA for 24 or 48 h, hepcidin mRNA levels were analyzed by real-time RT-PCR. Data are expressed as means ± SDs of five independent experiments. * *P* < 0.05 versus the untreated control. (**c**) Immunohistochemical analysis of hepcidin expression in liver tissues of mice, 7 days after treatment with 4-PBA. Scale bar = 50 μm. (**d**) DAB staining was measured with Image J program. Data are expressed as means ± SDs of five independent measurements. * *P* < 0.05 versus the untreated control.

**Figure 3 ijms-21-05516-f003:**
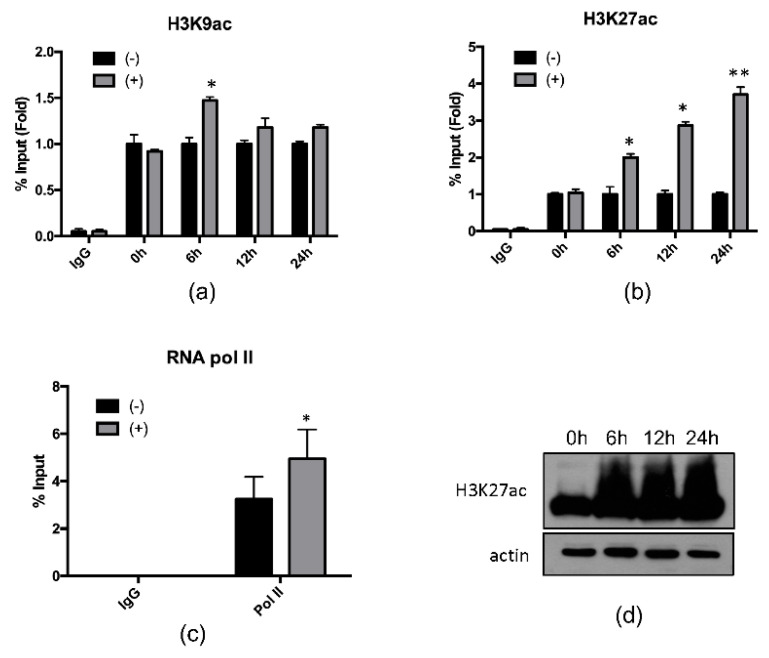
4-PBA upregulated hepcidin transcription through hepcidin locus acetylation. Hepcidin locus acetylation in Huh 7.5-Con1 cells treated with 4-PBA for the indicated times was analyzed by ChIP. Nonspecific IgG was used as a negative control in immunoprecipitation and subjected to qPCR. (**a**) H3K9ac, (**b**) H3K27ac, and (**c**) RNA pol II-binding DNAs were normalized to the % input method. (**d**) Whole-cell H3K27ac in Huh 7.5-Con1 cells treated with 1 mM 4-PBA for 24 h. Data are expressed as means ± SDs of five independent experiments. * *P* < 0.05, ** *P* < 0.01 versus untreated control.

**Figure 4 ijms-21-05516-f004:**
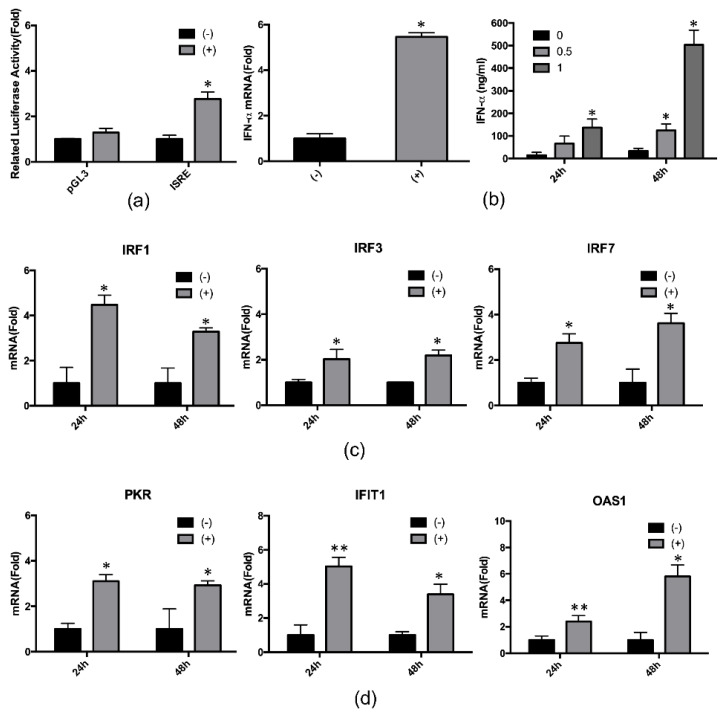
4-PBA enhanced IFN-related gene expression. (**a**) Huh 7.5-Con1 cells were transfected with the empty pGL3 vector or with ISRE constructs and then incubated with or without 1 mM 4-PBA for 24 h. Luciferase activity was expressed relative to that in untreated cells. (**b**–**d**) After treatment with 1 mM 4-PBA for 24 h, (**b**, **left**) *IFN-α*, (**c**, **left**) *IRF1*, (**c**, **middle**) *IRF3*, (**c**, **right**) *IRF7*, (**d**, **left**) *PKR*, (**d**, **middle**) *IFIT1*, and (**d**, **right**) *OAS1* mRNA levels were analyzed by real-time RT-PCR. (**b**, **right**) Huh 7.5-Con1 cells were incubated with 1 mM 4-PBA for 24 h. The medium was changed to serum-free DMEM during the 24-h incubation. After incubation, supernatants were harvested and analyzed using IFN-α-specific ELISAs. Data are expressed as means ± SDs of five independent experiments. * *P* < 0.05, ** *P* < 0.01 versus the untreated control.

**Figure 5 ijms-21-05516-f005:**
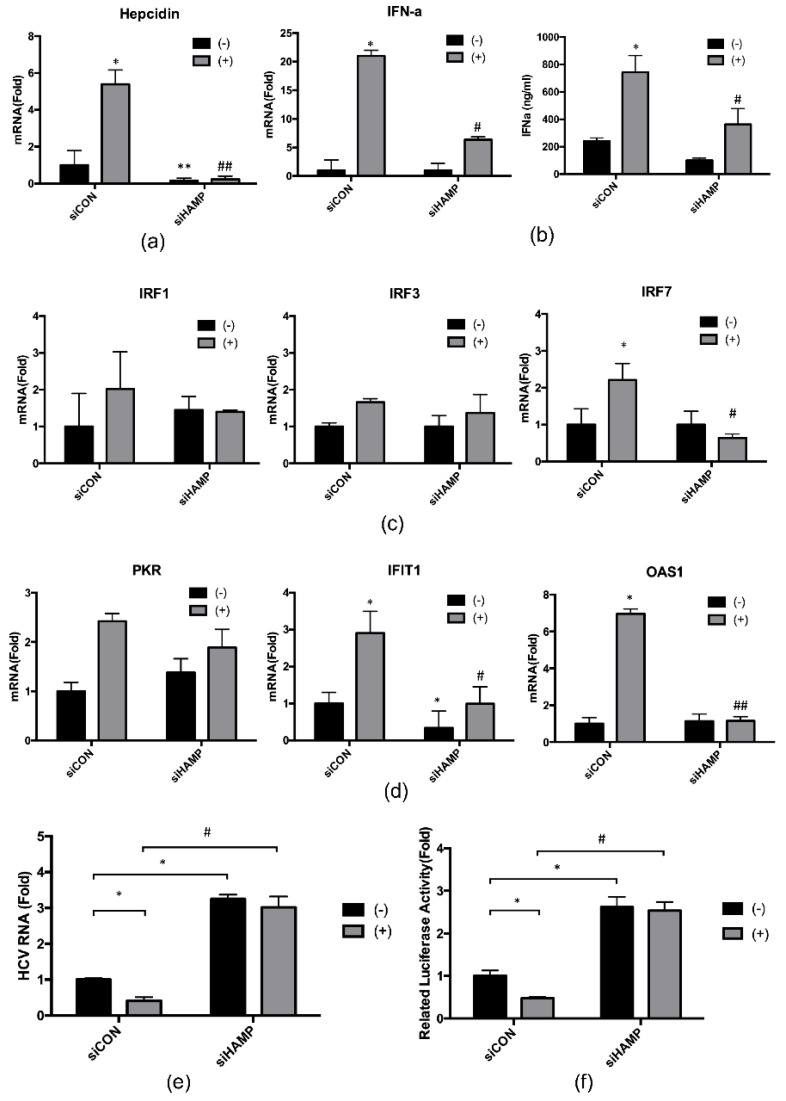
Hepcidin was involved in 4-PBA-mediated IFN-α signaling. Huh 7.5-Con 1 cells were transfected with control siRNA or hepcidin siRNA and treated with 1 mM 4-PBA for 24 h. After incubation, (**a**) hepcidin, (**b**, **left**) *IFN-α*, (**c**, **left**) *IRF1,* (**c**, **middle**) *IRF3*, (**c**, **right**) *IRF7*, (**d**, **left**) *PKR*, (**d**, **middle**) *IFIT1*, and (**d**, **right**) *OAS1* mRNA levels were analyzed by real-time RT-PCR. (**b**, **right**) siRNA-transfected Huh 7.5-Con1 cells were incubated with 1 mM 4-PBA for 24 h. The medium was changed to serum-free DMEM during the 24-h incubation. After incubation, supernatants were harvested and analyzed using IFN-α-specific ELISAs. HCV replication was measured by (**e**) qRT-PCR and (**f**) luciferase assays. Data are expressed as means ± SDs of five independent experiments. * *P* < 0.05, ** *P* < 0.01 versus the untreated siCON-transfected control; # *P* < 0.05, ## *P* < 0.01 versus 4-PBA-treated, siCON-transfected control; one-way ANOVA with Tukey’s multiple comparisons test.

**Figure 6 ijms-21-05516-f006:**
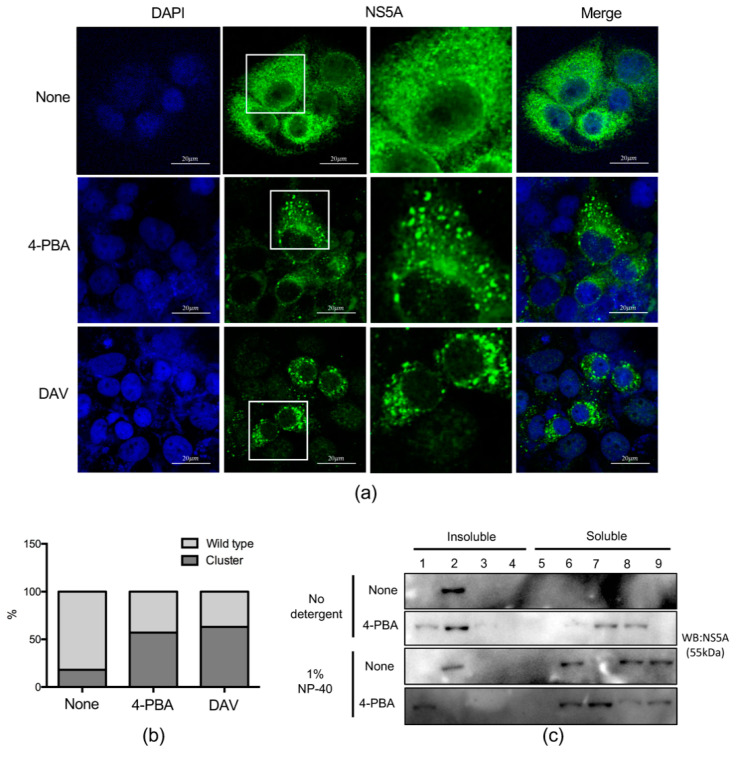
4-PBA altered membranous web formation. (**a**) Huh7.5-Con1 cells were treated with 4-PBA for 24 h, and NS5A (green) was detected by immunolabeling, with DAPI (blue) nuclear counterstaining. Scale bar = 20 µm (**b**) Immunolabeled NS5A cluster structures were detected using the particle analysis function of ImageJ software. Data are expressed as means ± SDs of five independent experiments. (**c**) Cells were harvested 24 h after treatment with 1 mM 4-PBA, and cell lysates were incubated on ice in the presence or absence of 1% NP-40 for 1 h and fractionated on a sucrose gradient. Each fraction was concentrated with trichloroacetic acid and analyzed by SDS-PAGE. NS5A was detected by immunoblotting. Lanes 1–4, insoluble fraction; lanes 5–9, soluble fraction.

**Figure 7 ijms-21-05516-f007:**
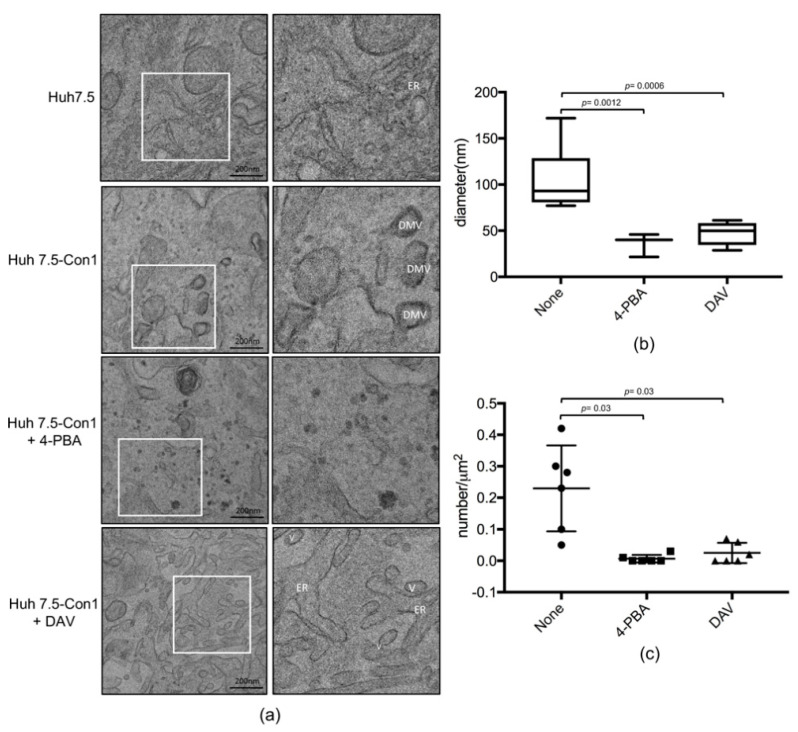
4-PBA inhibited DMV formation. (**a**) Electron micrographs of untreated Huh7.5 cells (upper panel) and Huh7.5-Con1 cells with or without 1 mM 4-PBA treatment for 24 h. Right panels show higher magnification images of areas enclosed by a dashed white square in the left panels. Cells treated with 2 μM DAV served as a positive control. DMVs were visualized by electron microscopy. DMV, double membrane vesicle; ER, endoplasmic reticulum; V, vesicle. Scale bar = 200 nm. Quantification of DMV diameter (**b**) and number of DMVs per square micrometer (**c**) in mock-treated cells versus cells treated with 1 mM 4-PBA and 2 μM DAV. The sizes of 90 DMVs per treatment group were measured using Digital Micrograph software. The surfaces of six cell profiles were measured, and the number of DMVs within each cell profile was determined using ImageJ software. *P* values shown in the graphs were calculated with unpaired Student’s *t*-tests.

**Figure 8 ijms-21-05516-f008:**
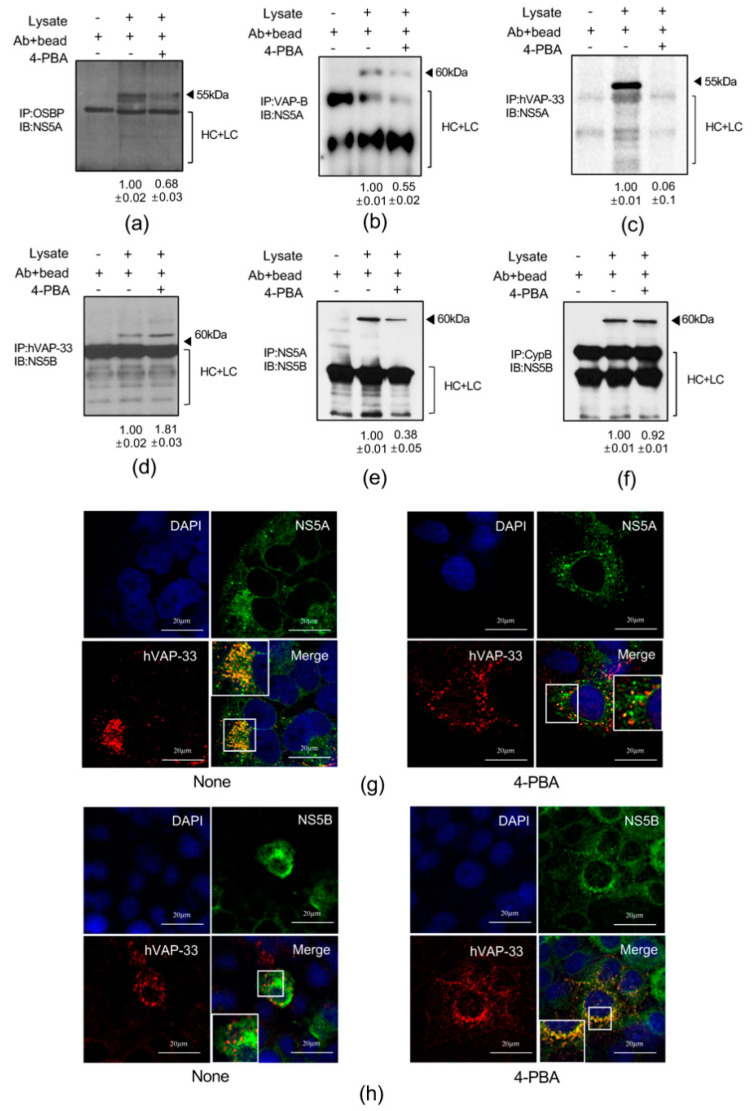
4-PBA inhibited replication complex interactions. (**a**–**c**) Immunoprecipitation was performed with Huh7.5-Con1 cells with or without 1 mM 4-PBA treatment for 24 h using antibodies against OSBP (**a**), VAP-B (**b**), or hVAP-33 (**c**), and precipitated protein complexes were probed with anti-NA5A antibodies. (**d**–**f**) Immunoprecipitation was performed with Huh7.5-Con1 cells treated with or without 1 mM 4-PBA for 24 h using antibodies against hVAP-33 (**d**), NS5A (**e**), or CypB (**f**), and precipitated protein complexes were probed with anti-NS5B antibodies. The arrowheads indicate antibodies used for immunoprecipitation. HC, antibody heavy chain; LC, antibody light chain. (**g**,**h**) Indirect immunofluorescence analysis of Huh7.5-Con1 cells with or without 1 mM 4-PBA treatment for 24 h using antibodies against NS5A (green) and hVAP-33 (red) (**g**) or NS5B (green) and hVAP-33 (red) (**h**). DAPI (blue) was used to visualize nuclei. Merged images of green and red signals are shown in panels. Scale bar = 20 μm.

**Figure 9 ijms-21-05516-f009:**
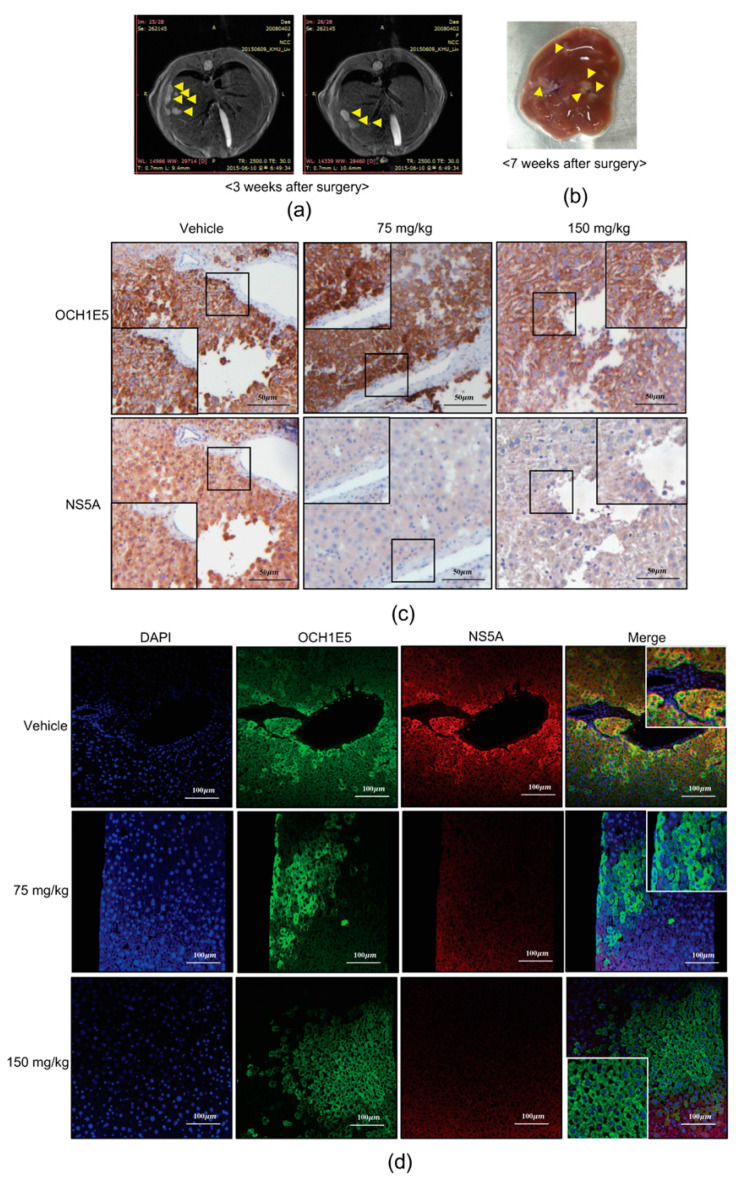
Antiviral effects of 4-PBA in an HCV mouse model. (**a**) MRI results showing the presence of cells injected into the spleen. MRI scans were performed on mice 3 weeks after intrasplenic injection. Animals were positioned on a probe in a head-first, prone position. T2-weighted images at 4.7 Tesla were obtained for 16 coronal sections at a slice thickness of 0.7 mm. (**b**) Morphology of mouse liver at 7 weeks. Yellow arrowheads indicate the formation of tumor-like clusters by injected Huh7.5-Con1 cells. (**c**,**d**) Immunohistochemical (**c**) and immunofluorescence analyses (**d**) of OCHIE5 and NS5A expression in liver tissues of the HCV mouse model 7 days after treatment with 4-PBA. Scale bar = (**c**) 20 μm, (**d**) 100 μm.

**Table 1 ijms-21-05516-t001:** List of primer couples generated for qRT-PCR.

Gene	Direction	Sequence
HCV	forward	CGGGAGAGCCATAGTGGTCTG CG
	reverse	CTCGCAAGCACCCTATCA GGCAGTA
b-actin	forward	AGGCTGTGCTGTCCCTCT
	reverse	TCCGGTGAGGAGGATGCG
hepcidin	forward	ACCAGAGCAAGCTCAAGACC
	reverse	CAGGGCAGGTAGGTTCTACG
IFN-a	forward	TTTCTCCTGCCTGAAGGACAG
	reverse	GCTCATGATTTCTGCTCTGACA
OAS1	forward	AGGTGGTAAAGGGTGGCTCC
	reverse	ACAACCAGGTCAGCGTCAGAT
IFIT1	forward	TGGCTAAGCAAAACCCTGCA
	reverse	TCTGGCCTTTCAGGTGTTTCAC
IRF1	forward	CTCACCAGGAACCAGAGGAA
	reverse	TGAGTGGTGTAACTGCTGTGG
IRF3	forward	ACCAGCCGTGGACCAAGAG
	reverse	TACCAAGGCCCTGAGGCAC
IRF7	forward	TGGTCCTGGTGAAGCTGGAA
	reverse	GATGTCGTCATAGAGGCTGTTGG
PKR	forward	TCTCTGGCGGTCTTCAGAAT
	reverse	ACTCCCTGCTTCTGACGGTA
hVAP-33	forward	GAAGACTACAGCACCTCGCC
	reverse	GCCTCTTTCCACACAGCTTC
Hepcidin_locus	forward	CTGTTTTCCCACAACAGGTG
	reverse	CTCAGTGCTCGGGTGTCTC

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
