# Peer review of "A Small Molecule, 4-Phenylbutyric Acid, Suppresses HCV Replication via Epigenetically Induced Hepatic Hepcidin"

_ijms, 2020, doi:10.3390/ijms21155516_

Round 1

Reviewer 1 Report

The Authors evaluated the effects of 4-Phenyl butyrate (4-PBA) on HCV replication. They demonstrated that the short-chain fatty acid 4-PBA inhibited HCV replication in vitro and in vivo.

This paper is a significant contribution to the scientific discussion about mechanisms of HCV RNA replication through induction of hepcidin expression. It has a good scientific quality (methods, statistical tests, quality of discussion).

Author Response

We are grateful for your careful reading of this manuscript, and we also very much appreciate generous comments. Thank you again.

Reviewer 2 Report

1. Inconsistency in many aspects:

a. Figures have a scale bar in 1e and 2b but not in 2c, 8g, and 8h.

b. Spacing between numbers and units, I.e., 150 mg/kg (line232) vs. fig. 2d etc.

2. Insertion fig. 1a in line 198 and change fig. 1, b into fig.1b, c (line 200), fig. 1c into fig. 1d, and fig. 1d to fig. 1e.

3. In fig. 7a, LD (lipid droplet) label is wrong. How come LD has a double layer of OS-philic feature.

Author Response

1. Inconsistency in many aspects:

a. Figures have a scale bar in 1e and 2b but not in 2c, 8g, and 8h.

Answer: We apologize for this mistake. In accordance with reviewer’s suggestion, we added the scale bar in Figure 2c, 8g, and 8h..

Please see the followings,

; Figure 2c

; Figure 8g,and 8h

b. Spacing between numbers and units, I.e., 150 mg/kg (line232) vs. fig. 2d etc.

Answer: In accordance with reviewer’s suggestion, we inserted a space between numbers and units.

Please see the followings,

; Fig 2c, and 2d

; Fig 9c, and 9d

2. Insertion fig.1a in line 198 and change fig.1, b into fig.1b, c (line 200), fig.1c into fig. 1d, and fig.1d to fig.1e.

Answer: We apologize for this mistake. In accordance with reviewer’s suggestion, we corrected the figure numbers.

Please see the followings,

; Page 5, line 202-203

; Page 5, line 203

; Page 6, line 206

; Page 6, line 209

3. In fig.7a, LD(Lipid droplet) label is wrong. How come LD a double layer of OS-philic feature.

Answer: We apologize for this mistake. In accordance with reviewer’s suggestion, we delated the wrong label.

Please see the following.

; Figure 7a.